# A cross-sectional study of barriers to cervical cancer screening uptake in Ghana: An application of the health belief model

Ama G. Ampofo[1,2,3☯]*, Afia D. Adumatta[1☯‡], Esther Owusu[1☯‡], Kofi Awuviry-Newton[4‡]

1 Department of Nursing, Garden City University College, Kumasi, Ghana, 2 Health Behaviour Research Collaborative, Priority Research Centre for Health Behaviour, School of Medicine and Public Health, University of Newcastle, Newcastle, Australia, 3 Hunter Medical Research Institute, New Lambton Heights, Australia, 4 Priority Research Centre for Generational Health and Ageing, Faculty of Health and Medicine, University of Newcastle, Newcastle, Australia

☯ These authors contributed equally to this work.
‡ These authors also contributed equally to this work.
* tyzy2009@hotmail.com

**Data Availability Statement:** Data may compromise the privacy of study participants and are therefore only available upon request. Due to these conditions, interested researchers can

## Abstract

### Background

The high incidence (32.9, age-standardized per 100,000) and mortality (23.0, age-standardized per 100,000) of cervical cancer (CC) in Ghana have been largely attributed to low screening uptake (0.8%). Although the low cost (Visual inspection with acetic acid) screening services available at various local health facilities screening uptake is meager.

### Objective

The purpose of the study is to determine the barriers influencing CC screening among women in the Ashanti Region of Ghana using the health belief model.

### Methods

A analytical cross-sectional study design was conducted between January and March 2019 at Kenyase, the Ashanti Region of Ghana. The study employed self-administered questionnaires were used to collect data from 200 women. Descriptive statistics were used to examine the differences in interest and non-interest in participating in CC screening on barriers affecting CC screening. Multivariable logistic regression was used to determine factors affecting CC screening at a significance level of p<0.05.

### Results

Unemployed women were less likely to have an interest in CC screening than those who were employed (adjustes odds ratio (aOR) = 0.005, 95%CI:0.001–0.041, p = 0.005). Women who were highly educated were 122 times very likely to be interested in CC screening than those with no or low formal education (aOR = 121.915 95%CI: 14.096–1054.469, p<0.001) and those who were unmarried were less likely to be interested in CC screening than those with those who were married (aOR = 0.124, 95%CI: 0.024–0.647, p = 0.013).

access to the underlying data by sending an e-mail request through the Garden City University College board (contact at christa@gcuc.edu.gh) to the data holder, the correspnding author, Ama G. Ampofo (ama.ampofo@uon.edu.au).

**Funding:** No specific funding for this work

**Competing interests:** The authors have declared that no competing interests exist.

Also, perceived threat, perceived benefits, perceived barriers and cues for action showed significant differences with interest in participating in screening with a P-values <0.003. The association was different for long waiting time, prioritizing early morning and late evening screening which showed no significant difference (P-value > 0.003).

## Conclusions

Married women, unemployed and those with no formal education are less likely to participate in CC screening. The study details significant barriers to cervical cancer screening uptake in Ghana. It is recommended that the Ghana health services should develop appropriate, culturally tailored educational materials to inform individuals with no formal education through health campaigns in schools, churches and communities to enhance CC screening uptake.

## Introduction

The incidence and mortality of cervical cancer in low-and lower-middle-income countries (LLMIC) continue to rise, including sub-Saharan African[1]. According to Globocan (2018), the age-standardized incidence and mortality rate per 100,000 women of cervical cancer (CC) in LLMIC countries is 28.8 and 22.1 making CC the leading cause of cancer-related mortality in these countries [1]. Ghana, one of the sub-Saharan countries, has a high age-standardized incidence and mortality rate of 32.9 and 23.0 per 100,000, respectively. These estimates make cervical cancer the most frequently diagnosed cancer after breast cancer (43.0, age-standardized per 100,000 women) and the leading cause of cancer deaths amongst women in Ghana [1].

Despite the burden of cervical cancer in sub-Saharan Africa few interventions such as mass media campaigns, special education and reminders have been used to create awareness of primary and secondary prevention of cervical cancer including screening [2]. No policies towards national screening programmes currently exist in these countries, and a similar situation is seen in Ghana. Thus if steps are not taken, it may lead to the estimated 5-year prevalence of CC (46.4, age standardized rate per 100,000) across all ages [1].

Although population-based cervical screening and guidelines have resulted in a substantial decline in the burden of CC in developed countries, the lack of screening programs contributes to the high risk of CC in sub-Saharan Africa [3–6]. In Ghana, the non-existent national screening program for CC [6], most of the time, is attributed to lack of funds, limited qualified personnel, and infrastructure for the development of widespread screening initiatives [7]. This deficit has resulted in CC screening test such as Pap smear or visual inspection with acetic acid (VIA) services being offered on local uncoordinated basis. In other words, it is considered as an opportunistic screening, where doctors request Pap smear and VIA for patients in clinics and hospitals either as part of general medical examination or for consultations related or unrelated to CC [8]. In addition to this, government and private hospitals have trained nurses and doctors who conduct CC screening, especially in various family planning units as a routine before the insertion of an intrauterine device (IUD). Furthermore, organized screening by benevolent organizations is done occasionally in various communities in Ghana. Despite the multiple avenues for CC screening, uptake for these services are very low [8–10] with participation rates lower than 3% in Ghana [11]. Therefore, the evidence regarding screening uptake in Ghana remains very scarce.

Given the evidence that CC screening has accounted for a significant decline in mortality in some countries, well-coordinated programs are vital in achieving such outcomes [12, 13]. High-income countries have seen a successful decrease in CC incidence and mortality, and

this has been greatly attributed to well patronized preventive screening programs [1]. For instance, the United Kingdom has experienced a significant 70% drop in mortality over the years, with 70–73% screening uptake between [14]. Also, a 60% reduction in mortality was seen in the Netherlands; where invitations for screening are sent out annually to eligible women to increase participation [15]. Moreover, since the introduction of Papanicolaou (Pap) smear-based screening in the United States, the incidence rate of CC among white and African-American women has dramatically decreased by over 50% and 70% respectively. [16].

Previous studies conducted in Ghana have established the inadequate knowledge about cervical cancer risk factors and screening amongst women, leads to low self-perception of risk on the disease and low patronage of screening services. For instance, Ebu et al. [17] reported a high number (93.6%) and (97.7%) of women in the central region of Ghana had inadequate knowledge of the risk factors and screening of cervical cancer respectively. On the contrary, 85.6% of college students in Ghana were aware of cervical cancer. Therefore the knowledge level of the disease amongst women in Ghana remains unclear [18].

Although the high incidence of cervical cancer exists, low CC screening uptake, little knowledge on cervical cancer and various factors may contribute to the low interest in CC screening in Ghana. According to the health belief model (HBM), one of the most widely used health behavioural models due to its ability to understand health behaviour, modifying factors such as sociodemographic characteristics, knowledge of the disease, perceptions threat (susceptibility and seriousness), benefits, barriers and self-efficacy of the illness, behaviour and its cues for action are likely to influence a particular behaviour [19]. In systematic reviews across countries including sub-Saharan regions, low awareness of CC, perceived barriers like cultural beliefs, perceived fear of screening procedure and adverse outcome, societal stigmatisation, embarrassment, and lack of spousal support have been largely attributed to CC screening [20–25]. This evidence suggests that literature perceived threat, benefits and self-efficacy of cervical cancer is limited in sub-Saharan countries.

In Ghana, a qualitative study investigated only the psychological barriers affecting CC screening [26]. Although Ebu et al. [17] assessed the barriers of CC screening in the Central Region of Ghana, this study could determine the associated factors with interesting participating in CC cancer such as sociodemographic characteristics (education, marital status and occupation). Moreover, the current study can find the association between CC screening interest and barriers to screening, applying thethe health belief model. Therefore the evidence regarding barriers to CC screening remains unclear.

To best of the authors' knowledge, no empirical study has investigated women's behaviours towards CC screening in the Kenyase, Ashanti region of Ghana given the evidence that CC incidence and mortality is more common in this area [27]. More to this evidence there exist a differential cultural belief is prevalent across regions in Ghana. This study sought to determine the 1) uptake and interest in CC screening, 2) knowledge level of CC, 3) barriers influencing interest in CC screening, 4) association between sociodemographic characteristics and CC screening uptake under the lens of the health belief model among women in Ashanti region of Ghana. The findings will provide an insights into factors affecting CC screening participation and appropriate ways of targeting educational interventions amongst women in Ghana.

## Materials and methods

### Study setting and population

The study was conducted at Kenyase in the Kwabre District in the Ashanti region in Ghana. Kenyase shares boundaries with Duase, Bouhban, and Bosore. Akans are the predominant ethnic group with a smaller proportion of the population originating from outside ethnic Akan.

Kenyase has quite a large number of people, which is about 5000. A significant activity of the area is petty trading. The study population consisted of reproductive women in Kenyase environs, including Market places and schools.

## Study design and period

The analytical cross-sectional study design was used to conduct the study between January and March 2019 in Kenyase, a suburb of the Ashanti Region of Ghana.

## Sampling size and sampling procedure

A sample size of 245 was calculated using 5% error, 95% confidence interval using the formula below [28]:

$$n = \frac{z^2 \times pq}{e^2}$$

Where;

n = minimum sample size.

z = confidence interval at 95%, 1.96.

p = estimated proportion of CC screening uptake in Ghana, 0.2.

q = 1-p.

e = margin of error at 5%.

The sample size (n) was calculated as follows;

$$n = \frac{1.96^2 \times 0.2(1 - 0.2)}{0.05^2}$$

n = 246

Convenience sampling where random shops and houses in the community under study were chosen and visited until the predetermined number of surveys were completed.

## Data collection procedure

Well-structured self-administered questionnaires, which required about 20 minutes, were developed. The description of the questionnaire was as follows:

The sociodemographic characteristics consisted of age, occupation, educational background, religion, marital status and number of children, family history of cancer. The participants' beliefs about CC and screening were measured by 37 items based on constructs adapted from the HBM, as shown in Fig 1. These included four subscales of HBM: perceived threat, perceived benefits, perceived barriers and cue for action. The items were arranged as section B was for awareness of CC with 12 response items. Section C–the perceived threat of CC referred to woman's perception about the chances of getting cervical cancer and beliefs concerning the severity of cervical cancer screening. Section D—perceived benefits of cervical cancer screening. Section E—perceived barriers to cervical cancer screening were categories into psychosocial barriers, socioeconomic barriers, and healthcare system barriers. Section G—cues for action. For each item, the respondents were asked to answer yes, no or I don't know.

The pre-test was conducted among fifteen (15) women within age considered in a different community not selected for the study to ensure the contents of the questionnaire was clear. The questions were piloted using a one-on-one interview with women by the study researchers. The Cronbach alpha, which is a measure of internal consistency ranging from 0.70 to 1.00, with high coefficients indicating high levels of reliability, was used to determine the validity and the reliability of the questionnaire. The Cronbach alpha of most of the questions was

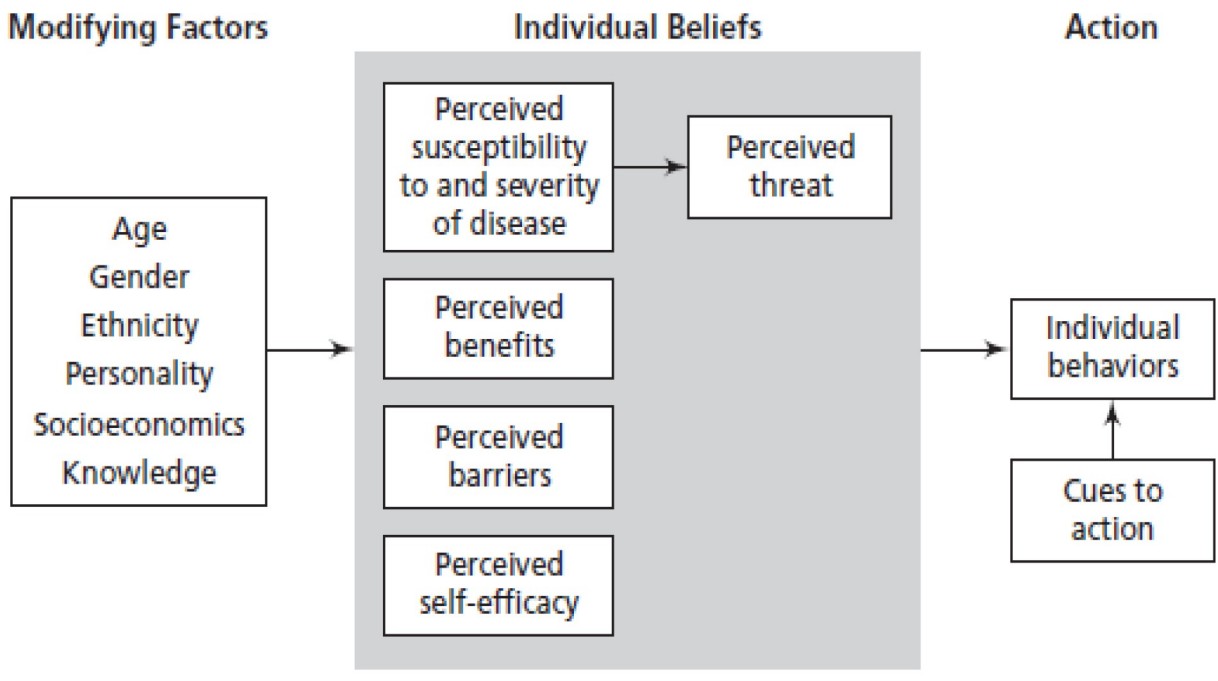

**Fig 1. Health belief model components and linkages.**

0.934. Appropriate changes were made to modify the questionnaire after the pilot study. The entire questionnaire was also available in English. The questionnaire was then personally delivered to the women in paper form by the research assistants. In situations where participants were illiterates, the researchers explained the questions to them in the local dialect (Twi).

## Data analysis

The data were analyzed using the Statistical Package for the Social Sciences (SPSS, IBM Corporation, Armonk, NY, USA) version 21.0. The results are presented with tables. The dependent variables were "the interest in participating in CC screening" and "have you been screened before" measured as "Yes" or "No" and presented as frequencies (percentages). With regards to assessing participants knowledge on cervical cancer, scoring was one '1' for each correct response and zero '0' for the wrong answer. The analysis of awareness and knowledge score was ranked as high if the total score is 6–9, fair if the overall score is 3–5 and low if the total score is less than 2. Pearson chi-square test was used to show the differences between proportions of individuals who were interested or not interested in participating in CC screening on the HBM subscales (perceived threat, perceived benefits, perceived barriers and cue for action). The significance level was set at p< 0.003 (Bonferroni's correction was used to adjust the significance level due to multiple comparisons). The independent variables were categorized as; the educational background was classified as high formal (tertiary graduates), low formal (primary, junior and senior high school), no formal education (illiterates). Marital status was married or unmarried (single, widowed, divorced and informal union). The number of children was none or more than or equal to one. Religion was categorized as Christian or Muslim, and family history was indicated as "Yes" or "No". Bivariate analysis was used to examine the relationship between the dependent variable (interest in participating in screening) and the explanatory variables. Variables that were not significant were dropped from the model

while those found to be significant were entered in multivariable logistic regression analysis. Crude odds ratio were adjusted a priori for age, family history, religious status, and number of children to identify factors influencing CC screening. Significance was set at p-value <0.05.

## Ethical approval

Approval was obtained from Garden City University College after the internal review board reviewed the proposal of the study. Also, an introductory letter was sent to the Unit Committee leader of the community and the queen mother of Kenyase. The proposal of the research was also reviewed by the committee leader of the community and consent was given to proceed with the study. Written consent of the participants was sought. Furthermore, participants had the freedom to participate or withdraw from the study at any time. Confidentiality and anonymity of respondents were ensured.

## Results

### Socio-demographic characteristics

There was 81% response rate since 200 questionnaires were returned. Majority of the respondents' 115 (57.5%) were between 26–40, and 155 (77.5%) of the respondents were employed. 119 (59.5%) of the respondents had low education, and 160 (80%) of the total respondents were Christians. 105 (52.5%) of the evaluated respondents were not married, and 151 (75.5.5%) of the respondents had more than or equal to one child or children while 78% had no family history of CC (see Table 1).

**Table 1. Respondents' sociodemographic characteristics amongst women in Kenyase, Ghana, N = 200.**

| Variable | Frequency N = 200 | Percentage (%) |
|---|---|---|
| **Age** | | |
| 15–25 | 54 | 27 |
| 26–40 | 115 | 57.5 |
| 41–50 | 31 | 15.5 |
| **Occupation** | | |
| employed | 155 | 77.5 |
| unemployed | 45 | 22.5 |
| **Educational background** | | |
| High formal education | 31 | 15.5 |
| Low formal education | 119 | 59.5 |
| No formal education | 50 | 25 |
| **Religion** | | |
| Christian | 160 | 80 |
| Muslim | 40 | 20 |
| **Marital status** | | |
| unmarried | 105 | 52.5 |
| married | 95 | 47.5 |
| **Number of children** | | |
| More than or equal to one | 151 | 75.5 |
| None | 49 | 24.5 |
| **Family history** | | |
| Yes | 44 | 22 |
| No | 156 | 78 |

## Interest and uptake in cervical cancer screening

Table 2 shows the interest and uptake CC screening amongst the respondents. More than half 163 (81.5%) had heard of CC screening. However, an overwhelming majority of 194 (97%) reported they had not been screened before. Out of the 6 (3%) who had been screened before, 4 (2%) had screened one year ago while 2 (1%) said they were screened not more than five years ago. 198 (99%) did not know how often they should go for screening. Interestingly, more than half 174 (87%) of the women evaluated said they were interested in participating in CC screening.

## Knowledge of cervical cancer

Table 3 presents knowledge of CC amongst respondents. Overall, 164 (82%) had scores below 5 for their level of knowledge of CC denoting inadequate understanding of cervical cancer. Only 36 (18%) of the respondents had adequate knowledge of cervical cancer. Almost all 190 (95%) of the women surveyed responded that CC was not a sexually transmitted disease and a little more than half 112 (56%) of the women surveyed indicated that CC could not be prevented. Also, majority 189 (94.5%) of the respondents mentioned it could not be prevented through vaccination of young girls.

Additionally, the majority of 143 (71.5%) of the respondents knew CC is curable in hospitals when diagnosed early. Most of the respondents had inadequate knowledge of the risk factors of cervical cancer. 60 (39%) of the respondents knew that multiple male sexual partners, while 78 (30%) knew a sexually transmitted virus caused CC. Surprisingly only 6 (3%) of them indicated HPV as a risk factor of CC, while 8 (4%) knew early sexual activity could lead to cervical cancer. Most of the respondent could not identify the signs and symptoms of CC. Less than half 71 (35.5%) of the respondents knew lower abdominal pains were part of the signs and symptoms of CC while few 25(12.5%), 11 (5.5%), 10 (5%) reported vaginal bleeding, pain in the genital during sexual intercourse and intermenstrual vaginal bleeding respectively as part of the signs and symptoms of CC.

**Table 2.  Interest and uptake cervical cancer screening amongst women in Kenyase, Ghana, N = 200.**

| Variable | Frequency N = 200 | Percentage (%) |
|---|---|---|
| **Have you heard of cervical cancer screening?** | | |
| Yes | 163 | 81.5 |
| No | 37 | 18.5 |
| **Have you been screened with cervical cancer before?** | | |
| Yes | 6 | 3 |
| No | 194 | 97 |
| **If Yes, Last time Screened** | | |
| No answer | 194 | 97 |
| A year ago | 4 | 2 |
| Not more than five years ago | 2 | 1 |
| **How often do you go for cervical cancer screening?** | | |
| Once a year | 2 | 1 |
| Don't know | 198 | 99 |
| Are you interested in participating in screening? | **Yes**: 174 | 87 |
| | **No**: 26 | 13 |
| **TOTAL** | **200** | **100** |

**Table 3. Knowledge of cervical cancer amongst women in Kenyase, Ghana, N = 200.**

| Variable | Frequency N = 200 | Percentage (%) |
|---|---|---|
| **Cervical cancer is a sexually transmitted disease** | | |
| Yes | 10 | 5 |
| No | 190 | 95 |
| **Cervical cancer is preventable** | | |
| Yes | 88 | 44 |
| No | 112 | 56 |
| **Cervical cancer is curable through vaccination** | | |
| Yes | 11 | 5.5 |
| No | 189 | 94.5 |
| **Cervical cancer is curable in hospitals when diagnosed early** | | |
| Yes | 143 | 71.5 |
| No | 57 | 28.5 |
| **Risk Factors of cervical cancer** | | |
| Early-onset of sexual activity | 8 | 4 |
| Infection with sexually transmitted infection | 60 | 30 |
| HPV | 6 | 3 |
| Multiple male sexual partners | 78 | 39 |
| Smoking cigarettes/tobacco | 12 | 6 |
| Grand multiparity | 17 | 8.5 |
| All the above | 19 | 9.5 |
| Don't know | 10 | 5 |
| **Signs and symptoms of cervical cancer** | | |
| Intermenstrual vaginal bleeding | 10 | 5 |
| Post-menopausal bleeding | 12 | 6 |
| Vaginal bleeding | 25 | 12.5 |
| Post-coital vaginal bleeding | 18 | 9 |
| Excessive vaginal discharge | 12 | 6 |
| Lower abdominal pain | 71 | 35.5 |
| Pain in the genital during sexual intercourse | 11 | 5.5 |
| All the above | 25 | 12.5 |
| Don't know | 16 | 8 |
| **Knowledge level on cervical cancer** | | |
| **High** | **Fair** | **Low** |
| **9–6** | **5–3** | **Less than 2** |
| 36 (18%) of the respondents scored high. | 74 (37%) of the respondents had a fair score. | 90 (45%) of the respondents had a low score. |

## Barriers to cervical cancer screening

Table 4 shows the responses to barriers that may affect CC screening among respondents. The main barriers identified by the respondents were individual beliefs and attitudes to screening, socioeconomic and healthcare system barriers. Majority of the respondents affirmed the fact that certain individual beliefs and attitudes may prevent them from participating in screening. For instance, a considerable number of the respondents 160 (80%) thought the screening was scary while about two thirds 149 (74.5%) of the respondents mentioned they were not susceptible to CC and thus will not screen. In addition to this, socioeconomic barriers such as lack of affordability and time may prevent respondents' likelihood of undergoing screening procedures. For instance, more than half of the respondents, 139 (69.5%) indicated screening was

**Table 4. Barriers to cervical cancer screening amongst women in Kenyase, Ghana, N = 200.**

| Variable | Yes, N (%) | No, N (%) | Don't know, N (%) |
|---|---|---|---|
| **Perceived threat** | | | |
| Are you afraid of bad diagnosis? | 147 (73.5) | 33 (16.5) | 20 (10) |
| Do you feel susceptible to cervical cancer? | 16 (8) | 149 (74.5) | 35(17.5) |
| Does the thought of cancer scare you? | 160 (80) | 27 (13.5) | 13(6.5) |
| **Perceived benefits** | | | |
| Is cervical cancer screening important? | 186 (93) | 0 (0) | 14 (7) |
| Do you believe cervical cancer can be cured? | 37 (18.5) | 87 (43.5) | 76 (38) |
| **Perceived barriers** | | | |
| *Psychosocial barriers* | | | |
| Does respondent's culture forbid cervical screening? | 0 (0) | 185 (92.5) | 15(7.5) |
| Is CC screening embarrassing? | 153 (76.5) | 35 (17.5) | 12(6) |
| Is cervical cancer screening painful? | 8 (4) | 27 (13.5) | 165 (82.5) |
| Does respondent's religion have anything against CC screening? | 11 (5.5) | 173 (86.5) | 16(8) |
| Is CC a curse from the gods? | 3 (1.5) | 182 (91) | 15 (7.5) |
| *Socioeconomic barriers* | | | |
| Is CC screening test affordable? | 47(23.5) | 14 (7) | 139 (69.5) |
| Can afford **USD 2–10** | 46 (23) | 0 (0) | 154 (77) |
| Is the transport system in the facility good? | 139 (69.5) | 29 (14.5) | 32 (16) |
| Do you have time for screening? | 23 (11.5) | 177 (88.5) | 0 (0) |
| *Healthcare system barriers* | | | |
| Are there long waiting time at health facility? | 172 (86) | 28 (14) | 0(0) |
| Does the respondent feel comfortable with male health personnel offering to screen? | 8 (4) | 157 (78.5) | 35 (17.5) |
| Does the respondent know any health facility offering CC screening? | 83 (41.5) | 110 (55) | 7(3.5) |
| Is it difficult to communicate with health personnel? | 133 (66.5) | 38 (19) | 29(14.5) |
| **Cues for action** | | | |
| Prioritizing early morning and late evening increase screening | 165 (78) | 7(3.5) | 37 (18.5) |
| Ensuring awareness of facility improve behavior for screening | 163 (81.5) | 14 (7) | 23 (11.5) |
| Will preferring female health personnel increase screening uptake? | 178 (89%) | 15 (7.5) | 7 (3.5) |

expensive, and 46(23%) could only afford between USD 2–10. Again, some aspects of the healthcare system are likely to pose as barriers to CC screening. For example, almost all the respondents 157 (78.5%) mentioned they did not like male health personnel offering screening services and 172 (86%) and 133 (66.5%) alluded that long waiting time at the health facility and communication respectively may prevent them from seeking CC screening services.

## The difference between respondents' interest in participating in screening and HBM constructs

Table 5 describes the difference between respondents' interest in participating in screening and HBM constructs. There was a significant difference between all the constructs and the respondents' interest in screening except long waiting time, CC can be cured and prioritizing early morning and late evening screening which showed no significant difference (P-value > 0.003).

## The relationship between respondents' sociodemographic characteristics and interest in participating in screening

Table 6 describes the sociodemographic predictors and interest in participating in CC screening using multivariate logistic regression analysis. Occupation (unemployed), educational background (high and no formal education) and marital status (married) were independent predictors of interest in participating in CC screening. Unemployed women were less likely to have an interest in CC screening than those who were employed (aOR = 0.005, 95%CI:0.001–0.041, p = 0.005). Women who were highly educated were 122 times very likely to be interested in CC screening than those with no or low formal education (aOR = 121.915 95%CI: 14.096–1054.469, p<0.001) and those who were unmarried were less likely to be interested in CC screening than those with those who were married (aOR = 0.124, 95%CI: 0.024–0.647, p = 0.013).

## Discussion

Although CC screening reduces the incidence and mortality rates of cervical cancer, women in developing countries have reported low screening uptake, especially in Ghana. The current

**Table 5. Chi-square analysis of respondents' interest in participating in screening on the perceived threat, perceived benefits, perceived barriers and cues of action amongst women in Kenyase, Ghana, N = 200.**

| Variable | Pearson chi-square | P-value |
|---|---|---|
| **Perceived threat of Cervical Cancer** | | |
| Thought of cancer is scary | 230.500 | <0.001* |
| Susceptibility to cervical cancer | 149.730 | <0.001* |
| Afraid of bad diagnosis | 20.123 | <0.001* |
| **Perceived benefits of cervical cancer screening** | | |
| Importance of cervical cancer screening | 185.663 | <0.001* |
| Cervical cancer be cured | 3.300 | 0.192 |
| **Perceived barriers of cervical cancer screening** | | |
| *Psychosocial barriers* | | |
| Cervical cancer screening is embarrassing | 14.874 | 0.005 |
| Culture forbid cervical screening | 92.529 | <0.001* |
| Religion against CC screening | 100.366 | <0.001* |
| CC is a curse from the gods | 94.472 | <0.001* |
| Cervical cancer screening painful | 16.580 | <0.001* |
| *Socioeconomic barriers* | | |
| Cervical cancer is screening expensive | 162.551 | <0.001* |
| Transport system | 130.575 | <0.001* |
| Have time for screening | 21.901 | <0.001* |
| *Healthcare system barriers* | | |
| Knowledge of health facility offering screening | 15.985 | 0.003* |
| Comfortable with male personnel | 17.594 | <0.001* |
| Long waiting time | 3.835 | 0.429 |
| Difficulties in communication | 121.585 | <0.001* |
| **Cues for action for cervical cancer screening** | | |
| Prioritizing early morning and late evening screening | 0.916 | 0.619 |
| Ensuring awareness of facility improve behavior for screening | 85.043 | <0.001* |
| Prefer female health personnel for screening | 95.667 | <0.001* |

*p-value statistically significant.

**Table 6. Sociodemographic factors predicting interest in participating in cervical cancer screening amongst women in Kenyase, Ghana, N = 200.**

| Variable | N | P-value | aOR (95% CI) [a] |
|---|---|---|---|
| **Occupation** | | | |
| Employed | 155 | Ref | |
| Unemployed | 45 | **0.005***  | 0.005 (0.001–0.041) |
| **Educational background** | | | |
| High formal education | 31 | <**0.001***  | 121.9 (14.095–1054.469) |
| Low formal education | 119 | Ref | |
| No formal education | 50 | 0.208 | 2.573 (0.591–11.211) |
| **Marital status** | | | |
| Unmarried | 105 | Ref | |
| Married | 95 | **0.013***  | 0.124 (0.024–0.647) |

*p-value statistically significant.

Adjusted for age, family history, religious status, and number of children.

aOR = adjusted odds ratio.

study revealed that only 3% of the respondents had ever patronized a CC screening test (either VIA or Pap test) even though there is high interest in participating in CC screening. This finding is consistent with a study conducted in which only 0.8% of the women and 8% of college students in the central region of Ghana had been screened before [17, 18]. Two systematic reviews reported similar findings of low screening uptake of 10%, 11%-19.9% and 14% in Tanzania, Ethiopia and Kenya, respectively [26, 29]. An explanation for these findings could be attributed to inadequate information about CC screening techniques and the unavailability of stimulating factors to enhance screening uptake. Thus national screening programs or routine screening during hospital visits could be adopted to improve screening rates.

Even though the study reported that majority of the respondents 194 (97%) had not been screened of CC before, a considerable number 163 (81.5%) had heard of screening, and more than 174 (87%) were interested in participating in screening. These findings were consistent with a study conducted in the capital of Ghana, Accra in 2009 amongst college students in which the majority of the respondents had heard of CC screening [30]. On the contrary, a study conducted in 2015 in the Central Region of Ghana found that a greater proportion (68.4%) of the participants had never heard of screening [17]. Likewise, studies in other developing countries like Zimbabwe, Ethiopia, and Nigeria [31–33]. A possible explanation for these differences in results could be attributed to improved CC education in Ghana across the years. Also, the fact that most of the women were interested in CC screening means that a national screening programme with intensive education and awareness is likely to see improved uptake.

Despite the high proportion of respondents knowing about CC screening, there was a low level of awareness of risk factors, signs and symptoms and preventive strategies. For instance, a notable finding is a fact that only 3% and 4% of the respondents could recognize HPV and first sexual intercourse as part of the risk factors of CC respectively [1, 34]. Previous studies in Ghana, Ethiopia, Nigeria, Kenya had similar findings [30, 33, 35–39]. Therefore, it was not surprising that majority of the respondents were not aware that CC ould be prevented through vaccination, which may result in low patronage of vaccination services. Hence, education on CC should be intensified since HPV has been highly associated with CC [1].

The key to effective screening programs depends primarimainly on an in-depth understanding of women and following clarification of beliefs and myths, which may lead to a reduction in barriers to screening despite high interest. The perceived threat, benefit, barriers and

cues for action may influence interest in participating in CC screening except for long waiting time, which showed no statistically significant difference in interest in participating in screening in the current study. However, a notable finding in the current study is the fact that psychosocial barriers, such as cultural taboos and beliefs and were not indicated to affect CC screening. In detail, the study revealed that fear of pain and the outcome of diagnosis influenced respondents' interest in screening. Studies conducted in Ghana and other sub-Saharan countries like Nigeria, Kenya have provided similar insights [17, 32, 40–43]. Also, the study indicated perceived lack of susceptibility and the importance of CC screening were significantly associated with interest in screening as well as high cost, transportation to the health facility and busy work schedules could affect interest in screening.

Another finding that prevented women from participating in screening is lack of knowledge on facilities offering screening services, male caregiver and difficulty in communication with the healthcare workers. These findings were affirmed in other studies conducted in Ghana and other sub-Saharan countries [32, 42–44]. Possible explanations to these outcomes could be because most women feel they do not experience signs and symptoms of the disease and thus may not find screening very important. Hence might result in a lack of time to attend or even seek locations where screening is ongoing or takes place.

The last HBM constructs, cues for action denotes a readiness to take action to improve interest in screening were found to include promoting awareness of screening facilities and using female screeners. The authors recommend that CC screening services should be incorporated into the national health insurance scheme as well as initiating a national screening program to improve coverage of CC screening services. Also, mobile screening sites could be set up in the communities, especially during market days, to enhance screening uptake.

The evidence concerning the likelihood of employment influencing the intention to screen is unclear. The current findings revealed that unemployed women were less likely than those employed to be interested in participating in CC screening, which is in line with studies conducted in Ghana and Nigeria. These studies reported a statistically significant association between employment and participating in CC screening services [45, 46]. Similarly, a study in Ethiopia said that women who were employed were four times more likely to utilize CC screening services [47]. However, amongst HIV patients, a study conducted in Ghana indicated that employment was not expected to affect the intention to have CC screening [48]. A possible explanation to this finding could be attributed to the fact that women who are employed have the financial ability to afford the cost of screening or perhaps they are keen on their health to be able to continue their work. Because CC services continue to be paid for, the occupation of women will play a significant role in seeking screening services as low socioeconomic status, including unemployed and part-time workers are less likely to patronise screening services.

The present finding revealed that women with high to no formal of education were more likely to show interest in CC screening. However, no formal education showed a statistically insignificant relationship with an interest in CC screening. This finding affirmed a study conducted in Ghana, in which respondents with a high level of education were more likely to screen [17]. Individuals with a higher level of education are more than 100 times likely to have an interest in CC screening. Thus, education could play a significant role in enhancing participation of CC screening activities. According to Ebu [48], educated women are placed in a position to understand health risk and so are more likely to engage in screening test. Education tends to change beliefs and unfavourable behaviours towards interventions put in place to enhance knowledge of health and illness [49]. Hence it is not surprising they have better chances of using maternal health services [50].

Furthermore, the finding of the current study suggests that married women are less likely to be interested in screening. Contrary to this finding, married women are two times more likely to

accept CC screening than those who are single in Tanzania [49]. An outcome which was not consistent with the current study. However, a study conducted in Ghana showed that marital status is not a statistically significant determinant of women' intention to have CC screening [48]. Thus the influence of marital status on CC screening appears to be inconsistent across studies. A possible explanation could be due to the concept of the different measurement across the studies as those who were in an informal union, or cohabiting were regarded as unmarried in the present study and other studies may classify them as married. Furthermore, the domineering power of men in African societies could prevent their partners from seeking CC services. This is because their partners influence health-seeking behaviours of women, and an example of this has been reported in situations whereby partners set up appointments with a physician [51].

Even though the current study has shown low screening uptake exists, and has provided insights into the factors affecting CC screening participation in Ghana, it's limitation cannot be overlooked. The major limitation is the less rigorous and biased sampling technique, convenience sampling, which makes generalizations impossible. Therefore, the likelihood of selecting more women who were interested in screening may be biased the findings. Thus future studies should use robust methods. This study has also contributed to the application of the health belief model to understand human behaviour by guiding the authors to identify various factors and barriers affecting cervical cancer screening Ghana.

## Conclusions

The study has documented that married women, unemployed and those with no formal education are less likely to participate in CC screening. Also, essential barriers relating to perceived lack of susceptibility, feeling of embarrassment, fear of wrong diagnosis and pains, feeling afraid, high cost, busy work schedule, lack of knowledge on screening facilities, the gender of screener and communication barriers affects women's interest in participating in screening. Hence the Ghana health services should develop appropriate, culturally tailored educational materials on CC screening to inform women particularly those with no formal education through social media, television, radio and community information systems to enhance uptake. Also, in the short term, the government of Ghana should make policies to incorporate cervical cancer screening to maternal health services such as family planning. In the long run, a national screening program should be instituted to enhance cervical screening uptake. Future studies should use larger sample sizes to make generalizations to improve policy decisions. Also, even though the current research identified various barriers that may affect interest in participating in CC screening using the HBM, the constructs of this model can be used in future studies to predict interest in CC screening among women in Ghana.

## Supporting information

**S1 File.**
(DOCX)

## Acknowledgments

A special thanks go to all the respondents who took part in this study, the Queen mother and opinion leaders at Kenyase for their approval and support during the study. All authors are appreciated for their time in improving the quality of this manuscript.

## Author Contributions

**Conceptualization:** Ama G. Ampofo, Afia D. Adumatta, Esther Owusu.

**Data curation:** Ama G. Ampofo, Afia D. Adumatta, Esther Owusu.

**Formal analysis:** Ama G. Ampofo, Afia D. Adumatta, Esther Owusu, Kofi Awuviry-Newton.

**Investigation:** Ama G. Ampofo.

**Methodology:** Ama G. Ampofo, Afia D. Adumatta, Esther Owusu.

**Project administration:** Afia D. Adumatta.

**Software:** Kofi Awuviry-Newton.

**Supervision:** Ama G. Ampofo.

**Validation:** Ama G. Ampofo, Afia D. Adumatta, Kofi Awuviry-Newton.

**Visualization:** Ama G. Ampofo, Afia D. Adumatta, Esther Owusu.

**Writing – original draft:** Ama G. Ampofo, Afia D. Adumatta, Esther Owusu, Kofi Awuviry-Newton.

**Writing – review & editing:** Ama G. Ampofo, Afia D. Adumatta, Esther Owusu, Kofi Awuviry-Newton.

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
