## [Editor Report · Decision Letter 0]

10 Oct 2019

PONE-D-19-23516

A Cross-Sectional Study of Barriers to Cervical Cancer Screening Uptake in Ghana: An application of the Health Belief Model

PLOS ONE

Dear Miss Ampofo,

Thank you for submitting your manuscript to PLOS ONE. After careful consideration, we feel that it has merit but does not fully meet PLOS ONE’s publication criteria as it currently stands. Therefore, we invite you to submit a revised version of the manuscript that addresses the points raised during the review process.

We would appreciate receiving your revised manuscript by Nov 24 2019 11:59PM. To enhance the reproducibility of your results, we recommend that if applicable you deposit your laboratory protocols in protocols.io, where a protocol can be assigned its own identifier (DOI) such that it can be cited independently in the future. For instructions see: http://journals.plos.org/plosone/s/submission-guidelines#loc-laboratory-protocols

We look forward to receiving your revised manuscript.

Kind regards,

Michael Johnson Mahande, PhD

Academic Editor

PLOS ONE

Journal Requirements:

Additional Editor Comments (if provided):

Section Comment, question, suggestion.

Abstract 1. On the introduction report high incidence and high mortality of Cervical Cancer with low screening uptake (needs to quantify)

2. Study period is missing in methods section

3. Reports descriptive Cross-sectional study design while multivariable logistic regression was done ( needs to change to analytical cross-sectional study design)

4. Missed recommendations and policy implications on Conclusion.

Introduction

Background 1. What is the burden of Cervical Cancer in Ghana (Incidence & mortality of Cervical Cancer)?

2. What are the interventions have been done in SA in recognition of the burden of cervical cancer? Are there any Strategic plans? Policies? (This will highlight any efforts which have been attempted to address the problem under study and outcomes of this effort)

3. Missed the gap (Why is the study being conducted?)

Literature review 4. Did not follow specific objectives (literature review should be reorganized to capture what is currently known about the proposed study outcomes/objectives, and it should provide available evidence about the topic under study from a global perspective to local)

a. Missed literature on Cervical Cancer Uptake

b. Missed literature on barriers to Cervical Cancer uptake

c. Factors influencing Cervical Cancer uptake applying the health belief model

Problem Statement 5. Problem under study is not stated clearly

6. Missed the knowledge gap (what information is lacking in the literature and what cannot be done because that information is not available?

Justification 7. Missed the benefits of study findings (to whom, why and how will the study findings benefit the beneficiaries)

8. Who are the beneficiaries of these findings?

Objectives 9. Objective number 2 is not clear “the study determines the association between sociodemographic characteristics and constructs under the lens of the health belief model”- misses the element of cervical cancer screening uptake.

Methodology 1. Missed the study duration (only mentions study was conducted on “March 2019”- it is not clear if this is the beginning or the end of the study).

2. What was the reason of doing convenience sampling technique?

3. How the Health Belief Model was introduced during data collection is not clear.

4. What was the criteria to conduct questionnaire pre-test among 15 women in the pilot study?

5. Prevalence of Cervical Cancer Screening uptake is >10%, were the alternative to logistic regression considered?

6. In the data analysis section SPSS is not cited correctly

7. Study design should be changed to analytical cross-sectional design as logistic regression was done

8. Why was classical analysis (chi-square) done after the regression analysis and what was the reason to do chi-square and logistic regression as per the objectives? It is not clear what method answers what objective

9. Were the priori confounders considered? And how were they controlled for?

10. Overall data analysis section should be revised and reorganized and should state clearly which method answers what objectives.

Results 1. It is important to mention the total sample size while reporting the results.

2. “N” showing total sample size should appear at the title of the table.

3. Results did not follow the specific objectives, makes it difficult to follow up.

4. What was being controlled for in the multivariable logistic regression analysis? Missed the bivariate analysis results

5. Overall results section should be re-organized per the specific objectives to enable the reader to follow through

Discussion 1. Discussion is also per the specific objectives; we do not discuss the background characteristics of study participants (Objective 0)

2. When mentioning study similarities and differences it is worth mentioning study areas and the study populations for clarity

3. Discussion did not follow specific objectives, missed the link between the study findings and the possible explanations basing on policies and interventions in place regarding Cervical Cancer screening.

Strengths and limitations 4. How is the current study able to improve Cervical Cancer screening uptake in Ghana? Please elaborate

5. How did the sampling design affect your study findings? Please elaborate more

6. What is the contribution of the current study to the scientific community?

Conclusion 7. Recommendations are not clearly stated (to whom, and what actions should be done, with what resources and under what timeframe)

8. What is the way forward basing on the study findings?
---

## [Author Response · Author response to Decision Letter 0]

23 Oct 2019

Section Comment, question, suggestion. Response 

Abstract 1. On the introduction report high incidence and high mortality of Cervical Cancer with low screening uptake (needs to quantify)

 The high incidence (32.9, age-standardized per 100,000) and mortality (23.0, age-standardized per 100,000)

 2. Study period is missing in methods section

 Study period included (January and March 2019)

 3. Reports descriptive Cross-sectional study design while multivariable logistic regression was done ( needs to change to analytical cross-sectional study design)

 Descriptive cross-sectional study design changed to analytical cross-sectional study design

 4. Missed recommendations and policy implications on Conclusion.

 Hence, the Ghana health services should develop appropriate, culturally tailored educational materials to inform individuals with no formal education through health campaigns in schools, churches and communities to enhance CC screening uptake. 

Introduction

Background 1. What is the burden of Cervical Cancer in Ghana (Incidence & mortality of Cervical Cancer)?

 Ghana, one of the sub-Saharan countries, has a high age-standardized incidence and mortality rate of 32.9 and 23.0 per 100,000 respectively. These estimates make cervical cancer the most frequently diagnosed cancer after breast cancer (43.0, age-standardized per 100,000 women) and the leading cause of cancer deaths amongst women in Ghana.

 2. What are the interventions have been done in SA in recognition of the burden of cervical cancer? Are there any Strategic plans? Policies? (This will highlight any efforts which have been attempted to address the problem under study and outcomes of this effort)

 No policies towards national screening programmes currently exists in these countries and a similar situation is seen in Ghana and thus if steps are not taken, it may lead to the estimated 5-year prevalence of CC (46.4, age standardized rate per 100,000) across all ages.

 3. Missed the gap (Why is the study being conducted?) Although population-based cervical screening and guidelines have resulted in a substantial decline in the burden of CC in developed countries, the lack of screening programs contributes to the high risk of CC in sub-Saharan Africa [3-6]. In Ghana, the non-existent national screening program for CC [6] have been mainly attributed to lack of funds, limited qualified personnel, and infrastructure for the development of widespread screening initiatives [7]. This has resulted in CC screening test such as, Pap smear or visual inspection with acetic acid (VIA) services being offered on local uncoordinated basis or as opportunistic screening, where doctors request Pap smear and VIA for patients in clinics and hospitals either as part of general medical examination or for consultations related or unrelated to CC [8]. Ina addition to this, government and private hospitals have trained nurses and doctors who conduct CC screening, especially in various family planning units as a routine before the insertion of an intrauterine device (IUD). Furthermore, organized screening by benevolent organizations is done occasionally in various communities in Ghana. In spite of the various avenues for CC screening, uptake for these services are very low [8-10] with participation rates lower than 3% in Ghana [11]. Therefore, the evidence regarding screening uptake in Ghana remains very scarce.

Literature review 1. Did not follow specific objectives (literature review should be reorganized to capture what is currently known about the proposed study outcomes/objectives, and it should provide available evidence about the topic under study from a global perspective to local)

a. Missed literature on Cervical Cancer Uptake

b. Missed literature on barriers to Cervical Cancer uptake

c. Factors influencing Cervical Cancer uptake applying the health belief model The literature review has been reorganized and it follows the study objective, and it provides global perspective now.

Problem Statement 1. Problem under study is not stated clearly

 Problem statement has been clearly stated

 2. Missed the knowledge gap (what information is lacking in the literature and what cannot be done because that information is not available? To best of the authors knowledge no empricial study has investigated the women’s behaviors towards CC screening in the Kenyase, Ashanti region of Ghana given the evidence that CC incidence and mortality is more common in this area [27], and the fact that differential cultural beliefs is prevalent across regions in Ghana.

Justification 1. Missed the benefits of study findings (to whom, why and how will the study findings benefit the beneficiaries)

 The findings of the provided good insights into factors affecting CC screening participation and appropriate ways of targeting educational interventions amongst women in Ghana.

 2. Who are the beneficiaries of these findings? Women in Ghana

Objectives 1. Objective number 2 is not clear “the study determines the association between sociodemographic characteristics and constructs under the lens of the health belief model”- misses the element of cervical cancer screening uptake. The objectives have been clearly stated now: This study sought to determine the 1) uptake and interest in CC screening, 2) knowledge level of CC, 3) barriers influencing interest in CC screening , 4) association between sociodemographic characteristics and CC screening uptake under the lens of the health belief model among women in Ashanti region of Ghana.

Methodology 1. Missed the study duration (only mentions study was conducted on “March 2019”- it is not clear if this is the beginning or the end of the study). January and March 2019

 2. What was the reason of doing convenience sampling technique? The sample was chosen as follows: random shops and houses in the community under study were chosen and visited until predetermined number of surveys were completed. The term convenience sampling was used because a structured sampling frame was not done due to the unstructured nature of the community, time and resource constraints.

 3. How the Health Belief Model was introduced during data collection is not clear. It was used in the questionnaire development and data analysis

 4. What was the criteria to conduct questionnaire pre-test among 15 women in the pilot study? The criteria for the pilot test was amongst women with the age considered in a different community not selected for the study.

 5. Prevalence of Cervical Cancer Screening uptake is >10%, were the alternative to logistic regression considered? Prevalence of cervical cancer screening uptake <10% and not used in logistic regression.

 6. In the data analysis section SPSS is not cited correctly Statistical Package for the Social Sciences (SPSS, IBM Corporation, Armonk, NY, USA) version 21.0.

 7. Study design should be changed to analytical cross-sectional design as logistic regression was done The analytical cross-sectional study design was used to conduct the study between January and March 2019 in Kenyase, a suburb of the Ashanti Region of Ghana.

 8. Why was classical analysis (chi-square) done after the regression analysis and what was the reason to do chi-square and logistic regression as per the objectives? It is not clear what method answers what objective Pearson chi-square test was used to show the differences between proportions of individuals who were interested or not interested in participating in CC screening in relation to the HBM constructs as subscales. This has been clearly stated now

 9. Were the priori confounders considered? And how were they controlled for? Yes, age, family history, religious status, and number of children were controlled for. However after controlling for each variable in the logistic regression model, those that were not significant at p<0.05.

 10. Overall data analysis section should be revised and reorganized and should state clearly which method answers what objectives. The data analysis has been reorganised as suggested. 

Results 1. It is important to mention the total sample size while reporting the results. The total results have been reported in each part of the results

 2. “N” showing total sample size should appear at the title of the table. N on the title has been stated

 3. Results did not follow the specific objectives, makes it difficult to follow up. Results have been reorganised according to specific objectives

 4. What was being controlled for in the multivariable logistic regression analysis? Missed the bivariate analysis results Age, family history, religious status, and number of children were controlled for. However after bivariate analysis these variables were dropped because they were not significant.

 5. Overall results section should be re-organized per the specific objectives to enable the reader to follow through The results section has been reorganised.

Discussion 1. Discussion is also per the specific objectives; we do not discuss the background characteristics of study participants (Objective 0) Background characteristics have been removed.

 2. When mentioning study similarities and differences it is worth mentioning study areas and the study populations for clarity Study areas and population has been stated

 3. Discussion did not follow specific objectives, missed the link between the study findings and the possible explanations basing on policies and interventions in place regarding Cervical Cancer screening. Discussion follow objectives and has been reorganised

Strengths and limitations 4. How is the current study able to improve Cervical Cancer screening uptake in Ghana? Please elaborate The current study has shown low screening uptake exists, and has provided insights into the factors affecting CC screening participation in Ghana.

 5. How did the sampling design affect your study findings? Please elaborate more The major limitation is the less rigorous and biased sampling technique, convenience sampling, which makes generalizations impossible. Therefore the likelihood of selecting more women who were interested in screening may be have biased the findings. Thus future studies should use robust methods

 6. What is the contribution of the current study to the scientific community? This study has also contributed to application of the health belief model to understand human behavior by guiding the authors to identify various factors and barriers affecting cervical cancer screening Ghana.

 7. How is the current study able to improve Cervical Cancer screening uptake in Ghana? Please elaborate The current study has shown low screening uptake exists, and has provided insights into the factors affecting CC screening participation in Ghana.

Conclusion 8. Recommendations are not clearly stated (to whom, and what actions should be done, with what resources and under what timeframe) The Ghana health services should develop appropriate, culturally tailored educational materials on CC screening to inform women particularly those with no formal education through social media, television, radio and community information systems to enhance uptake. Also, in the short term the government of Ghana should make policies to incorporate cervical cancer screening to maternal health services such as family planning. In the long term a national screening program should be instituted to enhance cervical screening uptake.

 9. What is the way forward basing on the study findings Future studies should use larger sample sizes to make generalizations to enhance policy decisions. Also, even though the current study identified various barriers that may affect interest in participating in CC screening using the HBM, the constructs of this model can used in future studies to predict interest in CC screening among women in Ghana.

---

## [Decision Letter · Decision Letter 1]

25 Mar 2020

A Cross-Sectional Study of Barriers to Cervical Cancer Screening Uptake in Ghana: An application of the Health Belief Model

PONE-D-19-23516R1

Dear Dr. Ampofo,

We are pleased to inform you that your manuscript has been judged scientifically suitable for publication and will be formally accepted for publication once it complies with all outstanding technical requirements.

With kind regards,

Ali Montazeri

Academic Editor

PLOS ONE

Additional Editor Comments (optional):

Reviewers' comments:

Reviewer's Responses to Questions

**Comments to the Author**

1. If the authors have adequately addressed your comments raised in a previous round of review and you feel that this manuscript is now acceptable for publication, you may indicate that here to bypass the “Comments to the Author” section, enter your conflict of interest statement in the “Confidential to Editor” section, and submit your "Accept" recommendation.

Reviewer #1: All comments have been addressed

2. Is the manuscript technically sound, and do the data support the conclusions?

Reviewer #1: Yes

3. Has the statistical analysis been performed appropriately and rigorously? 

Reviewer #1: Yes

4. Have the authors made all data underlying the findings in their manuscript fully available?

Reviewer #1: Yes

5. Is the manuscript presented in an intelligible fashion and written in standard English?

Reviewer #1: No

6. Review Comments to the Author

Reviewer #1: The authors have appropriately addressed reviewer comments.

Please correct few typographical and grammatical errors along the manuscript, pay attention to the abstract.

7. PLOS authors have the option to publish the peer review history of their article (what does this mean?). If published, this will include your full peer review and any attached files.

Reviewer #1: No

---

## [Editor Report · Acceptance letter]

21 Apr 2020

PONE-D-19-23516R1 

A Cross-Sectional Study of Barriers to Cervical Cancer Screening Uptake in Ghana: An application of the Health Belief Model 

Dear Dr. Ampofo:

I am pleased to inform you that your manuscript has been deemed suitable for publication in PLOS ONE. Congratulations! Your manuscript is now with our production department. 

With kind regards,

on behalf of

Professor Ali Montazeri 

Academic Editor

PLOS ONE